# Extracellular Vesicles in Lung Cancer: Implementation in Diagnosis and Therapeutic Perspectives

**DOI:** 10.3390/cancers16111967

**Published:** 2024-05-22

**Authors:** Anna Paola Carreca, Rosaria Tinnirello, Vitale Miceli, Antonio Galvano, Valerio Gristina, Lorena Incorvaia, Mariangela Pampalone, Simona Taverna, Gioacchin Iannolo

**Affiliations:** 1Ri.MED Foundation, 90127 Palermo, Italy; mpampalone@fondazionerimed.com; 2Department of Research, IRCCS ISMETT (Istituto Mediterraneo per i Trapianti e Terapie ad Alta Specializzazione), Via E. Tricomi 5, 90127 Palermo, Italy; rtinnirello@ismett.edu (R.T.); vmiceli@ismett.edu (V.M.); 3Department of Precision Medicine in Medical, Surgical and Critical Care, University of Palermo, 90133 Palermo, Italy; antonio.galvano@unipa.it (A.G.); valerio.gristina@unipa.it (V.G.); lorena.incorvaia@unipa.it (L.I.); 4Institute of Translational Pharmacology (IFT), National Research Council (CNR), 90146 Palermo, Italy; simona.taverna@cnr.it

**Keywords:** lung cancer, NSCLC, SCLC, EVs, BALF, liquid biopsy, personalized medicine, organ failure

## Abstract

**Simple Summary:**

Cell–cell communication mechanisms are gathering growing scientific interest, particularly in the context of cancer cells and the tumor microenvironment. Extracellular vesicles are gaining increased interest due to their relevance in tumor molecular characterization, classification, diagnosis, prognosis evaluation, and response to treatment. Many advances have been made in the clinical and therapeutic fields, exploiting increasingly precise biomolecular engineering strategies. This review aims to focus on the role of extracellular vesicles (EVs) as diagnostic and therapeutic tools in lung cancer.

**Abstract:**

Lung cancer represents the leading cause of cancer-related mortality worldwide, with around 1.8 million deaths in 2020. For this reason, there is an enormous interest in finding early diagnostic tools and novel therapeutic approaches, one of which is extracellular vesicles (EVs). EVs are nanoscale membranous particles that can carry proteins, lipids, and nucleic acids (DNA and RNA), mediating various biological processes, especially in cell–cell communication. As such, they represent an interesting biomarker for diagnostic analysis that can be performed easily by liquid biopsy. Moreover, their growing dataset shows promising results as drug delivery cargo. The aim of our work is to summarize the recent advances in and possible implications of EVs for early diagnosis and innovative therapies for lung cancer.

## 1. Introduction

Cancer is the leading cause of mortality globally [1], and a massive effort is being focused on finding novel therapeutic approaches and standardizing methods that can contribute to early neoplastic detection. Non-invasive techniques that do not involve radiation analysis represent a crucial goal. Among different tumors, the principal cause of death is lung cancer [1]. Lung cancer can be classified into two main histological subtypes: Small-Cell Lung Carcinoma (SCLC) and Non-Small Cell Lung Carcinoma (NSCLC), with a higher prevalence of NSCLC (about 80–85%) [2]. In the last decade, the high level of mortality due to lung cancer has prompted the onset of many multicenter studies seeking to improve early tumor detection by consolidated analysis (imaging by x-ray, PET, and PET/CT) and blood tests correlation. The 2004 COSMOS study (Continuous Observation of Smoking Subject) (ClinicalTrials.gov ID NCT01248806) enrolled more than 5000 asymptomatic smoker volunteers from the population because of their higher risk of developing lung cancer. Subjects were followed for 5 years with blood tests, spirometry, and annual low-dose spiral CT radiological examinations for nodules, alongside an evaluation of the correlation between COPD and lung cancer. Furthermore, many more studies comprising thousands of healthy patients have evaluated circulating biomarkers and radiomic analyses. For example, the CLEARLY study (Circulating and Imaging Biomarkers to Improve Lung Cancer EARLY Detection) (ClinicalTrials.gov ID NCT04323579), which started in 2018, is a multifactorial “bio-radiomic” protocol designed to detect early lung cancer in association with circulating biomarkers and radiomic data. Prognostic radiomic profiles for early detection have been correlated with molecular and cellular biomarkers such as microRNAs (miRNAs), proteins, circulating tumor cells (CTCs), and extracellular vesicles (EVs). EVs are involved in various processes, such as cell proliferation, differentiation, and the inflammatory response.

During the last ten years, circulating EVs have gained growing attention not only as biomarkers, but also for their ability to mediate cell–cell regulation and be manipulated for therapeutic purposes [3]. EV components have been implicated in many biological processes, and among them, a clear involvement in cancer invasion and metastasis has been observed [4]. Particularly noteworthy are the modulatory effects of EVs released from tumors and non-tumor cells such as mesenchymal stromal cells (MSCs) [5,6]. Many studies have been carried out to evaluate the effects and compositions of different EVs in tumor progression. The presence of regulatory messenger RNA (mRNA), which can modulate cancer cell proliferation, has been found within EVs [7]. Additionally, EV analysis has revealed the presence of controller proteins from neighboring cells [8], such as from the tumoral counterpart. Released EVs shuttle molecules involved in cell adhesion, migration, aggressiveness, and resistance to chemotherapeutic treatments [9]. The most remarkable molecules carried by EVs are miRNAs, which modulate multiple processes (growth, differentiation, apoptosis, migration, and drug/radioresistance) by their interaction with non-coding RNAs (ncRNAs), such as mRNAs, long non-coding RNAs (lncRNAs), and circular RNAs (circRNAs) [10]. Through these interactions, a single miRNA strand can control multiple genes, inhibiting their translation. This uniqueness gives relevance both to regulation processes and diagnosis and therapy. Engineering EVs with specific ncRNAs represents a promising outcome of the last few years, whereas the identification of an miRNA-specific signature from onset tumors still represents a challenging target. This review focuses on the role of EVs in diagnosis as components of liquid biopsy and in therapies for lung cancer, exploiting their use as theranostic agents. Despite many groups in the past describing the relationship between EVs and lung cancer, we hope that our work can help to suggest future diagnostic and therapeutic directions, improving their applications in fighting lung cancer [11,12,13].

## 2. Extracellular Vesicles

Extracellular vesicles (EVs) represent a crucial functional component of intercellular communication, acting as important mediators in both physiological and pathological processes in different organs and pathologies [14,15]. The classification of EVs reveals a complex landscape characterized by several factors. EVs were originally isolated from blood cells and showed significant variability in terms of their cellular origin, molecular content, size, and therapeutic efficacy [16,17]. Their classification based on size categorizes EVs into apoptotic bodies (1–5 µm), microvesicles (0.1–1 µm), and exosomes (30–150 nm) [18]. However, alternative classifications have been proposed, introducing considerations such as tissue of origin (e.g., prostasomes and oncosomes) and functional parameters [19]. EV proteins constitute a key aspect of their classification, reflecting both the cellular origin and the contents of the originating compartments. Exosomes (Exo) are generated by the endocytic pathway through the interaction between the endocytic vesicles and the endosomal sorting complex required for transport (ESCRT) system, and afterwards, they are released by the fusion of multivesicular bodies (MVBs) with the plasma membrane [20]. ESCRTs are involved in Exo production regulation also through the autophagy system. Autophagy-related genes (*Atg*) represent key factors for Exo release and their expression has been found to be deregulated in cancer cells, promoting proliferation and metastasis [21]. The complex network between autophagy and Exo trafficking includes many regulatory proteins and was recently revised by Zubkova and coworkers [22]. Conversely, microvesicles (MV) and apoptotic bodies arise directly from the plasma membrane [22]. In particular, MVs derive from membrane budding, whereas apoptotic bodies form from the blebbing of cells that undergo apoptosis. Cancer cells promote EV release to induce cancer development, proliferation, and metastasis. Among the EVs derived from cancer cells are oncosomes, which differ by size and composition from other EVs (Figure 1, Table 1).

Integral membrane proteins, specifically tetraspanins like CD9, CD63, and CD81, stand out as important markers. Furthermore, EVs may contain membrane and cytoskeletal proteins, lysosomal enzymes, cytokines, chemokines, antigen presentation-related proteins (MHC class I and II complexes), and nucleic acids such as DNA, mRNAs, and miRNAs, all of which contribute significantly to EV classification [23,36]. The existence of DNA in EVs demonstrated in the past decade adds an intriguing dimension to their molecular composition. DNA in EVs, different in type (single- or double-stranded, mitochondrial) and form (fragment or chromatin-bound), may aid in discriminating EVs based on their cell of origin [37,38]. However, due to a lack of sufficient biomarkers and an overlap in size range, it is difficult to discriminate between the various types of vesicles.

EVs function as messengers and can be involved in key physiological conditions such as coagulation, pregnancy, metabolism, immunity, and apoptosis [39,40,41,42,43]. Under pathological or stress conditions induced by various stimuli, EVs show dynamic responses by altering both their quantity and molecular composition [44,45,46,47,48]. These altered vesicles hold promise as prospective biomarkers for various diseases, serving as reservoirs for potentially dangerous molecules. The pivotal role of EVs extends to their involvement in neurodegenerative diseases [47], blood disorders [49], metabolic processes [50], and cancer progression [51], where they act as intercellular communicators between cells and distant organs. EVs carry functional biomolecules, such as mRNA, proteins, miRNA, and metabolites, and can deliver them to cells across short and long distances, using the blood as a transport medium. The growing interest in EVs as disease biomarkers is reflected in their detectability across various body fluids.

The innate targeting capacity of EVs has shown considerable potential in cancer therapy [52,53,54], where, to mitigate challenges such as rapid clearance, low uptake rates, and off-target effects, researchers have explored EV engineering strategies that involve the modification of the EV surface and internal cargos [55]. Recent studies have demonstrated that EV surface cargos significantly influence their uptake, providing a basis for engineering strategies. The surface markers, including integrins, CD63, and tetraspanin 8 [56,57], contribute to EV tropism and are susceptible to engineering to improve their uptake efficiency [58]. EVs’ potential in cancer therapy extends to artificial targeting strategies, where specific surface molecules are designed to bind to molecules expressed on the surface of the desired recipient cells. This approach includes receptor–ligand interactions, enzymatic modifications, and antigen–antibody combinations [55]. In particular, engineered EVs with ankyrin repeat proteins expressed on the surface of the cell membrane exhibited specific binding to HER2-positive breast cancer cells, showing the potential of the receptor–ligand interaction strategy [59]. Antibody-mediated strategies involve engineering EV surfaces with anti-CD3 and anti-EGFR antibodies, leading to the T-cell-mediated elimination of EGFR-positive cancer cells [60]. Enzymatic strategies using hyaluronidase on the EV surface aim to increase EV uptake by degrading the tumor extracellular matrix, improving permeability for both tumor-specific CD8 T cells and drugs in the tumor microenvironment [61].

Upon uptake, the EV cargo modulates the activity of recipient cells [62,63], and, in this context, EVs secreted by MSCs (MSC-EVs) are a promising therapeutic component of the MSC secretome. Most preclinical studies involving MSC-EV therapy are based on vesicles produced by MSCs [3,64,65]. Moreover, to potentiate the functional activity of MSC-EVs, the strategy of priming/preconditioning their cells of origin was explored by using chemicals, cytokines, and growth factors, as well as specific culture conditions [3,64,66,67,68,69]. For instance, human MSC-EVs produced after stimulation with dimethyloxaloylglycine further stimulated angiogenesis through the Akt/mTOR pathway to enhance bone healing [70]. Tumor necrosis factor-alpha (TNF-α) was able to prime MSCs and improve the bone regenerative properties of MSC-derived EVs, as evidenced by the increased proliferation and osteogenic differentiation of osteoblastic cells in vitro [71]. Furthermore, several studies explored the effects of inflammatory priming on MSC-EVs, revealing distinct EV functions compared to other priming conditions. For instance, it was recently demonstrated that EVs derived from IFN-γ-primed MSCs have improved immunomodulatory properties compared to the 3D culture priming of MSC-EVs, which instead showed enhanced angiogenic properties [66]. In this scenario, the yield, size, and surface marker composition of MSC-derived EVs exhibited substantial variations with various priming treatments, and it is intriguing to understand how the EV content and their beneficial properties can be modulated. These studies will no doubt lay the foundation for potential advancements in MSC-EV therapeutics.

## 3. EVs in Diagnosis 

While lung cancer represents, in most cases, a very inoperable disease with a low response to radiation therapy or chemotherapy and a low survival rate (with <17% for NSCLC and <7% for SCLC), the most important factor contributing to an increase in survival rate is early diagnosis and the selection of specific targeted therapeutic procedures. The identification of tumor characteristics based on molecular markers plays a key role in treatment effectiveness. Recently, a minimally invasive approach known as liquid biopsy was introduced, which involves sampling a small portion of body fluids to search for circulating tumor cells (CTCs), circulating proteins, and nucleic acids [72]. In this scenario, EVs, and particularly Exo, contain mediators influencing tumor progression as components of carcinogenesis that can help to identify and classify tumor onset and prevent its diffusion. 

Several methods can be used to isolate EVs, such as differential ultracentrifugation, size exclusion chromatography, gradient centrifugation, the co-precipitation method, and microfluidic devices [73]. Yet, this represents a major challenge for EV application, since the development of high-throughput methodologies to allow for the rapid isolation of EVs from many samples would enhance their use in diagnosis [74].

EVs are known to participate in intercellular communication, immune responses, metabolism, and tumor progression, as they are capable of horizontally transmitting a wide range of biomolecules to target cells, making them important biomarkers for diagnosis, as well as promising molecular carriers for targeted therapies. The information they carry can influence the behavior of target cells in multiple ways. In particular, they can act as indicators, transferring membrane proteins and receptors to target cells, or even altering their phenotype through the horizontal transfer of genetic information. It has been demonstrated that EVs can deliver not only proteins or lipids, but also miRNAs, other ncRNAs, and mRNAs [75]. The analysis of EV miRNA levels in lung cancer patients showed a significant difference compared to control samples, suggesting that circulating EV miRNAs might represent a useful screening tool [76]. Compared to other circulating biomarkers such as cell-free DNA (cfDNA) and CTC, EVs have the advantage of being more abundant and more stable, given their lipid layer, which also protects the transported cargo. These characteristics are very important in order to establish sensitive and easily repeatable protocols for the early diagnosis of disease. Their role is central in certain pathological phenomena; for instance, it is now widely demonstrated that a tumor cell can release more than 20,000 of these vesicles in 48 h [77], with a role in conditioning the tumor microenvironment (TME). The TME includes several components such as the extracellular matrix (ECM), endothelial cells, cancer-associated fibroblasts (CAFs), and a strong immune component such as tumor-associated macrophages (TAMs), natural killer cells (NK), and T and B lymphocytes. Sanchez and coworkers examined the involvement of EVs and their miRNA cargo in the TME, demonstrating how they stimulate the formation of the neointima by activating macrophages within the TME, thus generating a niche for inflammation [78]. The analysis of EVs can represent a low-impact source for lung cancer characterization; notably, it has been demonstrated that EVs derived from bronchoalveolar lavage fluid (BALF) liquid biopsy can be used proficiently for epidermal growth factor receptor (EGFR) genotyping and the evaluation of EGFR mutations [79]. This method, together with the digital droplet PCR (ddPCR) and next-generation sequencing (NGS) techniques, can allow for the stratification of patients for TKI treatment without invasive methods such as tissue biopsy [79]. In this way, it is possible to quantify (copies/mL) and identify, if present, variants relating to the mutated EGFR, perhaps due to targetable resistance mechanisms involved in resistance to cancer therapy [80]. In this regard, a prospective phase 2 study was carried out to promote EGFR genotyping for subsequent therapeutic interventions through the analysis of EV-BALF liquid biopsy obtained from advanced NSCLC patients [81]. The study, for the first time, established that this platform represents a valid tool for immediate genotyping and allows for rapid results for therapeutic initiation in advanced NSCLC patients [81]. Moving forward, genotyping the miRNA content in EVs has been widely investigated. A recent study evaluated, with low-dose computed tomography (LDCT), the presence of indeterminate pulmonary nodules (IPNs) in association with circulating EV miRNAs [82]. The NGS analysis demonstrated a specific miRNA signature associated with the patient’s prognostic survival [82]. Similarly, another study described an miRNA signature (hsa-miR-106b-3p, hsa-miR-125a-5p, hsa-miR-3615, and hsa-miR-450b-5p) from plasma-circulating EVs with the identification of early-stage lung cancer [83]. An analogous result was obtained with the RT-PCR analysis of six miRNAs (miR-7, miR-21, miR-126, Let-7a, miR-17, and miR-19) in EV-BALF. Despite the limited number of patients, the study suggested a correlation between the expression of the analyzed miRNAs and early-stage lung cancer [84]. High-throughput transcriptomic analyses allowed for the identification of circular RNAs (circRNAs), resulting from the back-splicing of pre-mRNA, among numerous RNA strands. Although first described in the early 1970s, circRNAs were, until very recently, regarded as byproducts of splicing without any important biological function. The main function of circRNAs is the inhibition of miRNAs. They act as miRNA sponges, establishing a complex and precise system in the interaction with RNA-binding proteins and in the regulation of gene expression [85]. Recently, circRNAs were found to be enriched and stable in cancer EVs, suggesting their potential use as cancer biomarkers or therapeutic targets. It has been supposed that EVs could represent a mechanism for the release of circRNAs [86,87].

Cancer patients show circRNA expression levels in the ratio of 2:1 vs healthy controls [88]. A valid example of the role of EVs in prognosis is given by the Hongya et al. study on circVMP1, which was found to be correlated with the progression of NSCLC and resistance to cisplatin therapy [89].

Indeed, there is much evidence for circRNAs being involved in promoting tumor migration, NSCLC development, resistance to therapies, and tumorigenesis, with different pathways of molecular interaction. Through the miR377-5p/NOVA2 axis, circ_007288 promotes the development of NSCLC [90], while circ_0000376 stimulates tumorigenesis and promotes drug resistance by positively modulating the action of KPNA4 and sponging miR1298-5p [91].

Circ_0020123 is particularly interesting for the multiple interaction pathways in which it is involved in lung cancer and appears to be capable of promoting cell proliferation and migration on tumor growth in vivo, acting on the THBS2/miR590-5p axis [92] and favoring cisplatin resistance in NSCLC cells by targeting miR-14-3p [93].

In the study conducted by Wei et al., circ_0020123 acts as a competitive endogenous RNA (ceRNA) to interact with miR-1283, thus promoting the expression levels of PDZD8, a cytoskeletal regulatory protein involved in tumor migration and proliferation [94], also involved in the LARP1/miR-330-5p tumor axis mechanism with the homonymous CircRNA (circ_PDZD8) [95] or suppressing tumor growth either if not expressed [96] or through sponging miR-1299, regulating HMGB3 [97]. Many studies on circRNA in lung cancer have demonstrated a repressive role in the disease. The relevance of circRNAs and their RNA splice variants for tumor progression and therapy response has been demonstrated in preclinical models [98]. Given the plethora of pathways in which circRNAs are involved, the use of a specific database is fundamental to shed light on the many possible pathways, and this is one of the objectives with which CircInteractome was born [99].

Recent studies have explored the role of circRNAs derived from the lung and carried by EVs [100], and most of them are focused on their expression and role in lung cancer [101] (Table 2).

In a pioneering work in this field, Zhu and coworkers identified the presence of circHIPK3 in lung cancer released EVs [102]. This circRNA has been proposed as a novel EV-derived biomarker for lung cancer, whose action is connected with miR-637 reduction and acts as a tumor suppressor on cellular migration, invasion, and proliferation in NSLC [102].

Moreover, it was reported that the circRNAs contained in EVs act as novel genetic information molecules, mediating the interactions between cancer cells and other cells of the TME and regulating key steps in cancer progression [10,103,104]. Nowadays, the use of EV-circRNAs as biomarkers for cancer diagnosis and prognosis shows various limitations for sample sizes and a lack of standardized evaluation systems, so further analysis will support their specific application as early diagnostic markers. 

On the other side, engineering strategies for EV-circRNAs could solve the limitations due to the size of circRNAs for efficient packaging and delivery systems, overcoming pharmacodynamics, pharmacokinetics, and safety considerations [105].

**Table 2 cancers-16-01967-t002:** circRNA effects on lung cancer.

CircRNA #	Function	Pathway	Reference
Circ_0012673	Promote cell proliferation	Sponge miR-22; upregulate ErbB3	[106]
Circ_0067934	Promote cell proliferation, migration, and invasion	Modulate markers of epithelial-to-mesenchymal transition (EMT)	[107]
Circ_007288	Promote cell proliferation	Sponge miR-377-5p/NOVA2	[90]
Circ_0000376	To induce resistance to cisplatin and promote tumorigenesis	Sponge miR-1298-5p/KPNA4	[91]
Circ_PDZD8	Promote cell proliferation	Sponge miR330-5p/LARP1	[95]
Circ_0072309	To promote tumor progression and invasion	Sponge miR607/FTO	[108]
Circ_ATAD1	Enhance cancer progression	Sponge miR-191-5p	[109]
Circ_0092887	Induce resistance to taxane	Sponge miR490-5p/UBE2T	[110]
Circ_0007385	Promote cell proliferation, migration, tumourigenesis, and invasion	Sponge miR-181	[111]
Circ_0013958	Promote cell proliferation and invasion and prevent apoptosis	Sponge miR-134/cyclin D1	[112]
Circ_0020123	Inhibit proliferation and invasion	Sponge miR1299/HMGB3	[97]
Circ_008305	Inhibit tumor metastasis	Sponge miR-429/miR-200b-3p/PTK2	[113]
Circ_CRIM1	Inhibit tumor metastasis and invasion	Sponge miR-93 and miR-182;	[114]
Circ_RNF13	Inhibit tumor proliferation and metastasis	Sponge miR-93-5p	[115]
CircSH3PXD2A	Inhibit tumor chemoresistance	miR-375-3p/YAP1	[116]

In addition to nucleic acid evaluation, recent progress in EV analysis has been implemented by looking at the protein content by proteomic profiling. Lung cancer EVs contain several tumor-associated proteins, such as EGFR, KRAS, inducer of extracellular matrix metalloproteinase, claudins, and RAB family proteins. In NSCLC, other proteins have been found such as exo markers like CD91, CD317, and EGFR. CD151, CD171, and tetraspanin 8 represent very reliable markers for lung cancer characterization and identification. Furthermore, METTL1 and the HIST family of proteins have been found to be overexpressed mostly in tumor samples [117]. Many studies are focusing on identifying the protein profiles of EVs from different stages and histologies of lung cancer, which is very important as a potential diagnostic tool [118,119]. A good example is given by Hoshino et al., who were able to characterize the complete proteomic profile of EVs from the plasma of 16 different cancer types and identified the proteins up- or down-regulated in cancer-associated EVs. Notably, the study revealed that cancer-derived proteins were not potential tumor tissue biomarkers and that approximately 50% of them arose from distant organs. Tumor-specific proteins were detected only in plasma, supporting the systemic nature of cancer and strengthening the potential use of EVs as liquid biopsy markers for early cancer diagnosis [117]. It has been reported that NSCLC-EVs shuttle specific proteins capable of inducing metastasis. Taverna et al. demonstrated that Amphiregulin, a ligand of EGFR contained in NSCLC-EVs, could induce metastasis, activating the EGFR pathway in pre-osteoclasts with an enhanced activity of proteolytic enzymes, leading to bone metastasis formation [120]. NSCLC EVs show an increased expression of *FAM3C*, a gene encoding for interleukin-like EMT inducer (ILEI). This results in an enhanced detection of FAM3C from lung tumor patients vs healthy subjects [121]. Furthermore, Du and coworkers identified that SCLC tumor-cell-derived EVs expressing PD-L1 play an important role in EVs and immune system crosstalk, suggesting a potential use of EV PD-L1 in the design of anticancer strategies [122]. From a prognostic point of view, the expression proteins of the A549 cell line (NSCLC) were analyzed before and after cisplatin treatment [123] by mass spectrometry (LC–MS/MS analysis). The results define a protein profile enriched for cholesterol metabolism pathway activation, indicating the role of EVs in lipogenesis activation and cell proliferation after chemotherapeutic treatment [123]. Nonetheless, a uniform consensus on protein markers from EVs is still missing for the restricted human sample datasets to drive interpretations of data analyses. To date, various resources have deposited the contents of EVs, especially regarding miRNAs, which can be consulted online: EVpedia [124,125] and Exocarta [126]. While the observation of new diagnostic information is strongly promoted, ctDNA represents an interesting target for liquid biopsy investigations in lung cancer detection [127]. However, the study of EVs and their protein cargo or CTCs has not yet entered clinical practice, and their application is limited to research studies (Table 3).

## 4. EVs in Lung Cancer Therapy

Until a few years ago, the most common lung cancer treatment was chemotherapy. Recent progress in oncology has prompted the use of immune-checkpoint monoclonal antibody blockades in association with chemotherapeutic treatment [133] or as a single agent, depending on PD-1 IHC expression. On the other hand, next-generation sequencing technologies allow for the identification of the most recurrent mutations in lung cancers, providing a unique tool for evaluating oncogene addiction and the role of targeted therapy. Some of the identified mutations include epidermal growth factor receptor (EGFR), where mutations occur in 15% of NSCLC adenocarcinoma cases [134]. This allows for the targeting of these tumors by specific tyrosine kinase inhibitors (TKIs) and/or monoclonal antibodies, as recommended by current guidelines [135]. Different TKIs have been employed in several clinical trials, which have demonstrated a positive effect on progression-free survival (PFS) and fewer side effects compared to standard chemotherapy (platinum) [136]. Unfortunately, many patients have shown resistance to the specific EGFR inhibitor treatment. To overcome this problem, TKI treatment can be associated with anti-EGFR monoclonal antibodies (cetuximab, necitumumab, and panitumumab), as supported by numerous clinical trials reviewed by Ciardiello and colleagues [137]. Another therapeutic target identified in lung cancers is anaplastic lymphoma kinase (ALK), whose translocation with the EML4 gene affects 5% of NSCLC patients [138]. Specific TKI inhibitors have been identified: crizotinib, second-generation ceritinib and alectinib, and the new-generation lorlatinib, recently preferred for resistance mutations [139]. Interestingly, crizotinib has also been employed as a treatment for NSCLC patients positive for ROS-1 chromosomal rearrangements with clinical signs similar to ALK mutations [140,141]. Similar to NSCLC cancers, some mutations have been identified in mainly SCLC patients. In particular, these alterations concern the suppressor genes TP53 and RB1 [142]. Despite their identification, SCLC tumors do not show targetable mutations, and recently, researchers have been focusing their attention on RB1 as a potential therapy target, as demonstrated by in vivo studies [143,144]. Innovative therapeutic approaches have been studied in the last few years, revealing that EVs play a relevant role in physiological and pathological conditions, such as cancer and cardiovascular and neurodegenerative diseases. Over the last ten years, EV research has focused on their potential application as therapeutic agents. As already underlined, EVs can carry molecules, particularly non-coding RNAs, influencing cancer growth, progression, metastasis, or drug resistance [145]. Therefore, non-coding RNA has gained importance as a therapeutic tool and has been employed in several clinical studies (Table 4). Among the ncRNAs, a pivotal role is played by miRNA, which can be easily carried and delivered by EVs or other vectors. Specifically, miR34 has been widely studied in different tumors. Recently two different phase I multicenter trials were conducted to study by dose escalation the safety, pharmacokinetics, and pharmacodynamics of an miR-34 mimic (MRX34), administered by liposomal injection in patients with melanoma (NCT02862145) and other selected solid tumors: primary liver cancer SCLC, NSCLC, lymphoma, melanoma, multiple myeloma, and renal cell carcinoma (NCT01829971). The melanoma trial was withdrawn due to high toxicity, and the other study on solid tumors showed stable disease (SD) in 6 out of 47 patients [146]. This study represented the first miRNA-based clinical trial on cancer [147]. The capacity of miR-34 to inhibit tumor growth has been demonstrated by various studies, and the ability of EVs to carry this miRNA and inhibit tumor growth in a paracrine way has been assessed [148]. EVs can be considered a peculiar vector for anti-cancer delivery systems due to their natural and advantageous properties, such as their high biocompatibility and limited systemic toxicity. Specific nanocarrier-targeted action can be improved by engineering and functionalizing their surface, for example, by inducing the expression of specific proteins on the EV membrane or through the loading of miRNA, which can be inserted exogenously on isolated EVs (electroporation, sonication, and RNA cholesterol conjugation), or indirectly by genetic modification of the donor cells before EV isolation (RNA transfection, RNA encoding plasmid transfection, and virus transfection) [145]. For example, EVs isolated from mesenchymal stem cells have been demonstrated to transfer miRNA efficiently in different kinds of tumors. This observation has raised the possibility of engineering cells such as MSCs for miR-34 delivery to inhibit tumor growth by EV release [149]. Notable for their ability to migrate towards inflammation or tumoral regions, MSCs have the peculiar characteristic of being able to be genetically modified, and when employed for this purpose, they act as living delivery vectors [150,151]. It was observed recently that engineered bone marrow MSCs (BMSCs) can deliver miR-193a, reducing the cisplatin resistance of NSCLCs by targeting leucine-rich repeat-containing protein 1 (LRRC1) [152]. In the same way, BMSC-derived EVs carrying miR-126-3p suppressed the viability, migration, and invasion of NSCLC cells by targeting protein tyrosine phosphatase non-receptor type 9 (PTPN9) [153]. Similarly, another group showed that engineered BMSCs with miR-598 inhibited cell proliferation, migration, and invasion in NSCLC. They demonstrated that miR-598-loaded EVs acted in lung cells by down-regulating Derlin-1, the zinc finger E-box-binding homeobox 2 (ZEB2), and also Thrombospondin-2 (THBS2), in this way inhibiting growth and metastasis [154]. The same effect was obtained with exosomal miR-338-3p through the inhibition of MAPK signaling, reducing the cell adhesion molecule L1-like protein (CHL1) activity and the subsequent down-regulation of NSCLC proliferation and apoptosis [155]. Engineered exosomes loaded with miR-449a selectively inhibit the growth of homologous NSCLC [156]. Among them, Zhou and colleagues focused their attention on miR-449-a, which affects the migration and invasion of human NSCLC cells. They isolated exosomes from A549 cells and engineered them (miR-449a exo) to allow for the transfer of this miRNA, thereby demonstrating its anti-tumor activity both in in vitro and in vivo models [156]. Similarly, another group used MDA-MB-231 breast cancer cells as a source of engineered lung-targeted exosomes with miRNA-126, which reduced proliferation and migration through the PTEN/PI3K/AKT pathway in A549 cells and an in vivo lung metastasis mouse model [157].

Besides their application as miRNA carriers, EVs have been used for tumor RNA interference (RNAi) therapy through siRNA targeted against specific oncogenes. For example, KRAS, whose mutations account for 90% of pancreatic cancers and 20–25% of lung adenocarcinomas, represents an area of great interest for tumor-targeted gene therapy. Recently, lipid nanoparticles carrying KRAS siRNAs reduced its expression in several lung cancer cell lines, including human (A549 and H441) and mouse (CMT-167 and Lacun3) cells, and proliferation was observed through colony-forming assays [158].

During the last few years, various approaches have been studied and pursued to employ EVs as therapeutic applications or targets in lung cancer. It is well known that the EVs released by tumor cells can promote the spread and diffusion of the tumor and also counteract the immune response by inhibiting CD-positive T cells with anti-tumor functions [159] or favoring immune escape, attenuating cytotoxic CD8+ T cells through the expression of PD-L1, considered as a target for monoclonal therapy in NSCLC patients [160]. Because of these characteristics, EVs have been considered as target therapeutic strategies. Some pharmacological agents act on EV trafficking or lipid membrane metabolism and are extremely important for membrane fluidity and, as a consequence, for EV shedding/release. For example, GW4869 inhibits the membrane-neutral sphingomyelinase (nSMase) and exosome/EV biogenesis; it has been tested in PC9 lung adenocarcinoma cells, counteracting the antagonistic effects of gefitinib and cisplatin, which are widely used for NSCLC patient treatment [161].

Among the numerous molecular partners involved in membrane trafficking is Rab27A, a protein expressed in numerous cell types, including A549, which could regulate EV release. One research group demonstrated that specific shRNA against Rab27A carries a lower release of EVs and a reduction in tumor growth in an in vitro model of human lung adenocarcinoma cells [162].

Considering the impact of EVs on immune escape, over the years, clinical trials have been undertaken to apply them as a cancer vaccine [163,164,165,166,167] The EVs released by tumor cells proficiently trigger anti-tumor immunity; for example, in a study focused on EVs in vitro isolated from 3LL lung tumor cells, the activation of dendritic cells and T cells after being subjected to heat stress was induced through EV inflammatory chemokine contents [163]. Similarly, dendritic cells release vesicles (termed dexosomes) that have been demonstrated to prime T cells and present antigens to T CD8+ and CD4+ cells [168,169]. These cells and their secretome are of great scientific interest; indeed, dendritic cells were tested as autologous vaccinations in a clinical trial involving NSCLC patients, providing interesting immunologic responses [164]. A phase I clinical trial demonstrated the tolerance of engineering dexosomes with MAGE antigens in NSCLC patients’ MAGE+ [167]. These dexosomes were also used in a phase II trial on NSCLC patients, resulting in the stabilization of 32% of the recruited patients [166].

In a similar way to miRNA delivery, researchers are attempting to use EVs for drug/chemotherapy delivery. EVs loaded with paclitaxel were administered to a metastatic mouse model of NSCLC [170]. In particular, this research group demonstrated that exosomes efficiently vehicle the paclitaxel [171] and subsequently improved the formulation of these exosomes, demonstrating that this new delivery system exerts a higher ability to reach cancer cells with a better therapeutic effect [170]. Recently, exosomes isolated from M1 macrophages were evaluated as a drug vehicle for cisplatin, both in in vitro (Lewis lung cancer cells) and in vivo mouse models. The study demonstrated that the exosomes from M1 macrophages as chemotherapy carriers improved the anti-lung cancer effect of cisplatin and induced tumor cell death; specifically, in vitro experiments demonstrated the involvement of apoptosis through Bax and Caspase-3 [172]. In another in vitro study with two NSCLC cell lines (H1299 and A549), researchers used exosomes loaded with gold nanoparticles conjugated with doxorubicin, obtaining a greater particle uptake by target cells and drug release and more specific cytotoxicity with fewer side effects [173].

**Table 4 cancers-16-01967-t004:** Therapeutic in vitro and in vivo application of EVs in lung cancers.

Target/Study Models	Subject	Description	Reference
(Advanced) NSCLC	Vaccination trial with tumor antigen-loaded dendritic cell-derived exosomes	Maintenance immunotherapy in 47 patients with dexosomes to improve their PFS.	NCT01159288
Solid tumors: primary liver cancer, SCLC, lymphoma, melanoma, multiple myeloma, renal cell carcinoma, NSCLC	Multicenter phase I study of MRX34, microRNA miR-RX34 liposomal injection	Phase I, open-label, multicenter, dose escalation study to investigate the safety, pharmacokinetics, and pharmacodynamics of the micro ribonucleic acid (microRNA) MRX34 in patients with unresectable primary liver cancer or advanced or metastatic cancer with or without liver involvement or hematologic malignancies.	NCT01829971 [147]
(Advanced) NSCLC	Phase I study of dexosome immunotherapy	Phase I study to evaluate safety and efficacy of autologous dexosomes loaded with tumor antigens (MAGE-A3, -A4, -A10, and MAGE-3DPO4), administered in 4 doses. Measurement of the immunologic responses and monitoring the clinical outcomes in 13 patients at different stages.	[167]
H1299 and A549 (NSCLC)	Nanosomes carrying doxorubicin anticancer activity against human lung cancer cells	In vitro analysis of gold nanoparticles (GNPs) loaded with doxorubicin to evaluate the release kinetics and the cytotoxic activity.	[173]
Mice injected with B16F10 cells to produce lung metastasis	EVs melanoma gold conjugated nanoparticle targeting lung tumors	The study provided an application system where exosomes isolated from cancer cells incorporated gold nanoparticles were tested in a mouse model to improve targeting system in metastatic foci.	[174]
In vitro: murine carcinoma cell line (3LL-M27); in vivo: mouse model with pulmonary metastases	Paclitaxel-loaded EVs against cancer cells	In vitro and in vivo study aims to introduce a new formulation for Paclitaxel distribution through exosomes (PTX-exo, fom RAW 264.7 cell line), providing high stability in tumor environment and a better effectiveness in vivo murine model.	[171]
In vitro: A549 and H1299 (NSCLC); In vivo: mouse model with lung cancer xenograft	Celastrol EVs formulation against lung cancer	Study focused on the effect of the natural compound celastrol loaded into exosomes, a new delivery system improved efficacy and reduced dose toxicity.	[175]
In vitro: A549 and H1299 (NSCLC); In vivo: nude mice with xenograft	Anthocyanidins EVs against multiple cancer types	The study aimed to obtain a nano-formulation of the natural derived compound, anthos, with exosomes. Exosomes enhanced the anti-proliferative and anti-inflammatory activity of anthos (vs the free compound) and the therapeutic affect toward lung cancer.	[176]
Nude mice with lung tumor xenografts	Milk-derived exosomes for oral delivery of paclitaxel	A study on chemotherapeutic paclitaxel delivery through exosomes in a formulation for oral administration, which exhibited greater therapeutic efficacy and lower systemic toxicity.	[177]

## 5. Conclusions and Remarks

The potential applications of EVs in therapeutic and diagnostic approaches are far from being fully achieved. Over the last decade, the EV cancer field has experienced significant advancements that have fundamentally changed our understanding of intercellular communication and cancer biology. 

However, a deeper knowledge of EV’s role in lung cancer is crucial in order to define biomarkers for prognosis and diagnosis, as well as to develop new therapeutic strategies for such deadly tumors [1]. So, to transfer this knowledge from bench to bedside, other studies need to be conducted to clarify and confirm the potential role of EVs in lung cancer and beyond. Tumor heterogeneity, in particular looking at EGFR mutations, is currently under investigation to further correlate cellular modifications with therapeutic response [81].

Their utility as delivery vehicles for various drugs, proteins, and nucleic acids has been evaluated by many laboratories. Their lipid composition contributes to their stability in body fluids and provides, at the same time, valid support for their cellular delivery by cell membrane fusion [178]. Moreover, the immunological properties of MSC offer a unique tool for EV secretion, combining their specific transfer ability aptitude (drugs, nucleic acids, and proteins) with immunomodulatory pharmacological effects [179] or new therapeutic approaches in numerous diseases, including lung cancer (Table 4). Despite MSCs’ natural tropism against tumors, which can represent a valid site-specific EV throughput tool, dendritic cell-derived exosomes can support the targeted tumor delivery of EVs and represent a promising example of vaccination due to their immunostimulatory capability (NCT01159288). On the other hand, from a diagnostic point of view and given the important role for cancer biology, the use of circulating EVs has gained a growing interest primarily for their availability. Conversely, one of the main challenges is represented from EVs’ origin, because their release is not exclusively related to the disease but can arise from any tissue. A wider analysis of EVs’ composition can support fast stratification and early detection. In this regard, a substantial analysis of EV circRNA signatures can identify lung-cancer-regulated miRNA [100,102]. Furthermore, a proteomic analysis of EV content offers the opportunity to acquire more information about EV biology and identify new biomarkers, contributing to early diagnosis and the design of valid treatments [180] (Figure 2). There are many difficulties and limitations, but the multi-omics approach has a very bright future and will undoubtedly provide much more information on these nano-sized biological entities. Despite numerous studies on experimental models and various pathologies, there are still many points that can be improved, for example, identifying cellular sources safe for immunogenicity and sources that can guarantee significant quantities, as well as trying to introduce standardized procedures to improve the workflow throughput. We hope that groundbreaking tests on the diagnostic and prognostic meaning of EV evaluation can draw new routine procedures for dissecting tumor heterogeneity and narrowing therapeutic intervention protocols. 

Last, but not least, scientists must investigate EVs’ structure deeply to maximize their engineering and applications as carrier systems (Figure 2). Another area to be further explored is related to their turnover. Studies have already focused on their release inhibition, and, considering the importance of the uptake step, it could be interesting to try to selectively reduce uptake mechanisms, although the pathways involved are numerous [181,182].

## Figures and Tables

**Figure 1 cancers-16-01967-f001:**
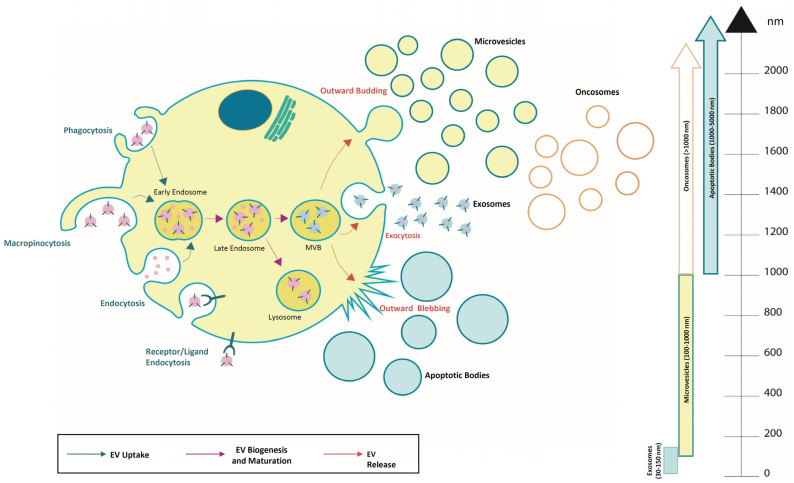
Overview of extracellular vesicle subtypes and their uptake, biogenesis, and release. They are classified into different sub-classes and are generated through the endosomal pathway, released as exosomes (30–150 nm), microvesicles (0.1–1 μm), apoptotic bodies (1–5 µm), and oncosomes (>1 µm).

**Figure 2 cancers-16-01967-f002:**
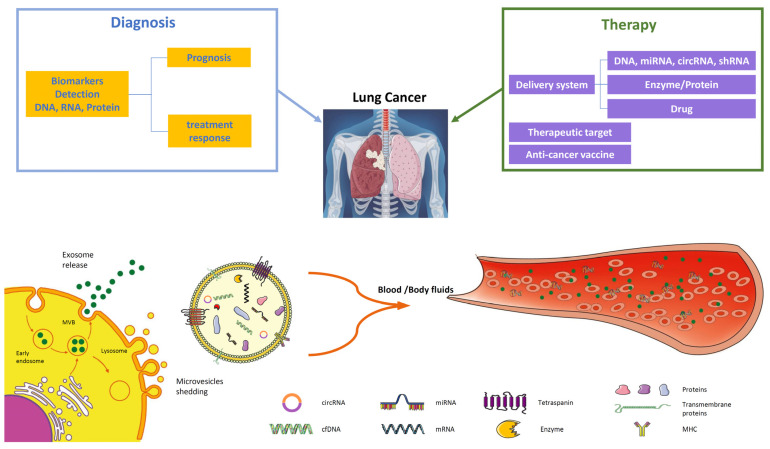
EVs in lung cancer diagnosis and therapy. EVs are important players in intercellular communication, released through the endosomal pathway by the plasma membrane as exosomes (30–150 nm), microvesicles (0.1–1 μm), and apoptotic bodies (1–5 µm). Tumor-derived EVs are good candidates for liquid biopsy since they contain many components such as tumor-derived DNA, mRNA, miRNAs, and proteins. Their analysis from plasma or body fluids (BALF) offers significant information about tumor diagnosis through biomarkers crucial for early detection or prognosis and treatment response. The potential application of EV in therapy comprises their application in targeted therapy through the delivery of specific miRNAs, drug delivery of chemotherapy agents, or their employment as anti-cancer vaccines.

**Table 1 cancers-16-01967-t001:** EV classification.

Characteristics of Extracellular Vesicles (EVs) Subtypes				
EV Subtypes	Origin	Markers	Cargo	Reference
Exosomes	MVBs fuse with plasma membrane	CD63, CD81, CD9, HSP60, HSP70, Alix, TSG101	Genomic DNA, mRNA, miRNA, circRNA, lncRNA, MHC class I and II	[23,24,25]
Microvesicles	Outward budding of plasma membrane	Anneximìn A1, Integrins, CD62, CD40 ligand	mRNA, miRNA, circRNA, lncRNA, Lipids, Adesion proteins	[26,27,28]
Oncosomes	Exclusively shed by cancer cells;Outward budding of plasma membrane	CAV-1, Keratin 18, ARF6, GAPDH	Genomic DNA, mRNA, miRNA, circRNA, lncRNA, MHC calss I and II	[29,30,31,32]
Apoptotic bodies	Outward blebbing from cells in apoptosis	Caspase 3, Annexin V, CD63, CD81	miRNA, mRNA, Fragmented DNA, Histones	[33,34,35]

**Table 3 cancers-16-01967-t003:** Diagnostic application of EVs from different body fluids in lung cancer.

Disease	Body Fluid Samples Source	Description	Reference
Lung Cancer	BALF	LC-MS analysis of proteome profile. DNMT3B protein complex as potential therapeutic target.	[128]
Early-Stage Lung Adenocarcinoma	BALF	Quantitative analysis of miRNAs with diagnostic value. miR-126 and Let-7a possible diagnostic biomarkers: higher levels in lung adenocarcinoma than in control subjects.	[84]
Early-Stage Lung Adenocarcinoma/Invasive Stage Lung Adenocarcinoma	Plasma	A signature drawn up with four miRNAs (hsa-miR-106b-3p, hsa-miR-125a-5p, hsa-miR-3615, and hsa-miR-450b-5p) for early diagnosis.	[83]
Advanced-Stage Lung Adenocarcinoma	BALF	Next-Generation Sequencing (NGS) of EV DNA content to identify genetic alterations, suitable for a clinical approach.	[129]
(Advanced) NSCLC	BALF	EGFR mutation analysis on BALF EVs as method more accurate, specific and rapid than cfDNA evaluation.	[79]
(Advanced) NSCLC	Plasma and BALF	BALF EV DNA analysis as alternative diagnostic method in accordance with tissue biopsy and greater efficiency for detecting the p.T790 M mutation in the patients resistant to EGFR-TKIs.	[130]
(Advanced) NSCLC	BALF	A phase 2 study on BALF EV as platform for EGFR genotyping and rapid therapeutic intervention.	[81]
Adenocarcinoma, Squamous Cell Carcinoma, NSCLC	Bronchial Washing	Detection of *EGFR* mutation and evaluation of its prognostic value.	[131]
Early-Stage Malignant Pleural Mesothelioma (MPM) vs Benign Conditions and Metastatic Adenocarcinomas	Pleural Effusions	Characterization of surface marker or proteins differentially expressed as diagnostic markers.	[132]
Indeterminate Pulmonary Nodules (IPNs)	Plasma	CircEV-miR profile as a molecular model to distinguish the benign and malignant IPNs. miR-30c-5p, miR-30e-5p, miR-500a-3p, miR-125a-5p, and miR-99a-5p: five miRNAs differentially expressed and associated to an overall survival.	[82] Chinese Clinical Trials: ChiCTR1800019877

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
