# Peer review of "Extracellular Vesicles in Lung Cancer: Implementation in Diagnosis and Therapeutic Perspectives"

_cancers, 2024, doi:10.3390/cancers16111967_

Round 1

Reviewer 1 Report

Comments and Suggestions for Authors

 Carreca et al. address the relevance of extracellular vesicles (EVs) in achieving diagnosis and therapeutic avenues in lung cancer. Emerging evidences on EVs are compelling to better understand the heterogeneity of various cancer types including lung cancer. There are suggestions that can help in the better impact of this paper.

1. A section on intracellular and extracellular tumor heterogeneity may be included for the broader aspects of EVs. Then followed by the EVs as a component of extracellular heterogeneity in lung cancer.

2. The authors should make strong rationale behind the selection of lung cancer over other cancer types and implications of EVs.

3. An additional discussion may be included on the EVs and extra-chromosomal circular DNAs and their association with lung cancer.

4. A future experimental avenues may be presented that can address the detection of EVs heterogeneity in lung cancer.

5. A discussion on the EVs and epigenetic regulation may be added.

Comments on the Quality of English Language

Moderate

Author Response

Carreca et al. address the relevance of extracellular vesicles (EVs) in achieving diagnosis and therapeutic avenues in lung cancer. Emerging evidences on EVs are compelling to better understand the heterogeneity of various cancer types including lung cancer. There are suggestions that can help in the better impact of this paper.

  1. A section on intracellular and extracellular tumor heterogeneity may be included for the broader aspects of EVs. Then followed by the EVs as a component of extracellular heterogeneity in lung cancer.

Thank you for the suggestion the text has been modified in accordance to the requests.

  1. The authors should make strong rationale behind the selection of lung cancer over other cancer types and implications of EVs.

The text was implemented as requested. Moreover, we respectfully draw attention to the introduction where we indicated the reason for investigating on lung cancer: “Among different tumors, the principal cause of death is lung cancer [1]”. As well the choose of EVs as main topic is due to the reason that: “Released EVs shuttle molecules involved in cell adhesion, migration, aggressiveness, and resistance to chemotherapeutic treatments [9]”.

  1. An additional discussion may be included on the EVs and extra-chromosomal circular DNAs and their association with lung cancer.

The requested discussion was inserted in association with table 2 on circRNA

  1. A future experimental avenues may be presented that can address the detection of EVs heterogeneity in lung cancer.

The text has been modified following the request.

  1. A discussion on the EVs and epigenetic regulation may be added.

The discussion has been inserted following the request.

Reviewer 2 Report

Comments and Suggestions for Authors

Even though the similar topic have been published before (Extracellular Vesicles in Lung Cancer: Prospects for Diagnostic and Therapeutic Applications, Cancers 2021, 13(18), 4604; https://doi.org/10.3390/cancers13184604; The roles of small extracellular vesicles in lung cancer: Molecular pathology, mechanisms, diagnostics, and therapeutics, BBA - Reviews on Cancer 1876 (2021) 188539, https://doi.org/10.1016/j.bbcan.2021.188539), but I noticed the authors renewed many fresh papers from 2021 (~64 papers, 20% of in total citation), so it’s still novel. And the structure of the manuscript is reasonable.

Here’re some suggestions.

Major

1.     Figures are too less. For part “2. Extracellular vesicles”, a scheme should be added; for part “3. EVs in diagnosis” and part “4. EVs in lung cancer therapy”, some figures should be inserted to make the manuscript richer.

2.     I’d suggest adding a table with the title “Diagnosis application of EVs in lung cancer” in part “2. Extracellular vesicles”, or use this kind of table to replace original Table 1 since the main point should be how EVs diagnose lung cancer, not how circRNA affect lung cancer.

Minor

1.     I’d suggest the author cited the previous review papers I listed in the beginning and indicate what’s the novelty of your paper compared to these two similar topic papers.

2.     Full name should be listed when the abbreviation appears first like EV, while “Dex” represents “dendritic cell-derived exosomes” looks weird.

3.     “In vitro” and “in vivo” should be italic.

4.     Line 88, the authors mentioned “categorizes 87 EVs into apoptotic bodies (1–5 µm), microparticles (0.1–1 µm), and exosomes (30–150 nm)”, but on line 493, it’s “exosomes (30-200 nm), microvesicles (200-1000 nm), and apoptotic bodies (1-5 µm).”, please make the content matched well in your manuscript.

5.     In table 2., only the line of “Subject” shows upper letter for each word.

Author Response

Even though the similar topic have been published before (Extracellular Vesicles in Lung Cancer: Prospects for Diagnostic and Therapeutic Applications, Cancers 2021, 13(18), 4604; https://doi.org/10.3390/cancers13184604; The roles of small extracellular vesicles in lung cancer: Molecular pathology, mechanisms, diagnostics, and therapeutics, BBA - Reviews on Cancer 1876 (2021) 188539, https://doi.org/10.1016/j.bbcan.2021.188539), but I noticed the authors renewed many fresh papers from 2021 (~64 papers, 20% of in total citation), so it’s still novel. And the structure of the manuscript is reasonable.

We thank the reviewer for the positive statement

Here’re some suggestions.

Major

  1. Figures are too less. For part “2. Extracellular vesicles”, a scheme should be added; for part “3. EVs in diagnosis” and part “4. EVs in lung cancer therapy”, some figures should be inserted to make the manuscript richer.

As indicated, we inserted a scheme on Extracellular vesicles (Figure 1)(Part #2). At the same time, we modified Figure 2 to underline EVs role in diagnosis and potential applications in therapy (Part #3 and #4).

  1. I’d suggest adding a table with the title “Diagnosis application of EVs in lung cancer” in part “2. Extracellular vesicles”, or use this kind of table to replace original Table 1 since the main point should be how EVs diagnose lung cancer, not how circRNA affect lung cancer.

We agree with your point: table1 was replaced. However, to accomplish another reviewer’s requests the table summarizing circRNA effects was maintained to shed light on the potential future use in diagnosis or therapy of these ncRNAs.

Minor

  1. I’d suggest the author cited the previous review papers I listed in the beginning and indicate what’s the novelty of your paper compared to these two similar topic papers.

Thank you for pointing out that, the text has been implemented (PMID: 34572829, PMID: 33892051, PMID 36829523)

  1. Full name should be listed when the abbreviation appears first like EV, while “Dex” represents “dendritic cell-derived exosomes” looks weird.

We thank the reviewer for noticing, we modified the text according to the suggestion.

  1. “In vitro” and “in vivo” should be italic.

We apologize and we totally agree, but MDPI rules do not consent the italic font in “in vitro” or “in vivo”

  1. Line 88, the authors mentioned “categorizes 87 EVs into apoptotic bodies (1–5 µm), microparticles (0.1–1 µm), and exosomes (30–150 nm)”, but on line 493, it’s “exosomes (30-200 nm), microvesicles (200-1000 nm), and apoptotic bodies (1-5 µm).”, please make the content matched well in your manuscript.

We thank the reviewer for noticing, we matched the description in: EVs into apoptotic bodies (1–5 µm), microparticles (0.1–1 µm), and exosomes (30–150 nm).

  1. In table 2., only the line of “Subject” shows upper letter for each word.

We thank the reviewer for noticing, the table was modified.

Round 2

Reviewer 1 Report

Comments and Suggestions for Authors

The authors have incorporated substatial changes in the manuscript in accordance with the previous suggestions. 

Author Response

The authors have incorporated substatial changes in the manuscript in accordance with the previous suggestions.

We thank the reviewer for the positive statement

Reviewer 2 Report

Comments and Suggestions for Authors

Everything looks good except this one:

I didn’t see any modification in figure 2, even though the authors mentioned “we modified Figure 2 to underline EVs role in diagnosis and potential applications in therapy”.

Author Response

Everything looks good except this one:

I didn’t see any modification in figure 2, even though the authors mentioned “we modified Figure 2 to underline EVs role in diagnosis and potential applications in therapy”.

We apologize for this point, in the revised version, the figure was slightly modified in comparison to the first submission, in this new version we hope that the figure will fit with the reviewer’s requests:

Major

  1. Figures are too less. For part “2. Extracellular vesicles”, a scheme should be added; for part “3. EVs in diagnosis” and part “4. EVs in lung cancer therapy”, some figures should be inserted to make the manuscript richer.

As indicated, we inserted a scheme on Extracellular vesicles (Figure 1)(Part #2). At the same time, we modified the Figure 2 to underline EVs role in diagnosis and potential applications in therapy (Part #3 and #4).

Round 3

Reviewer 2 Report

Comments and Suggestions for Authors

What I mean is supplement more typical figures from cited important and classic literature and get copyright, not only rotate the direction from the original figure.

Author Response

As requested, we modified Figure 2 following reviewer indications.